# Effect of Matrix Size Reduction on Textural Information in Clinical Magnetic Resonance Imaging

**DOI:** 10.3390/jcm11092526

**Published:** 2022-04-30

**Authors:** Michał Strzelecki, Adam Piórkowski, Rafał Obuchowicz

**Affiliations:** 1Institute of Electronics, Lodz University of Technology, 90-924 Lodz, Poland; michal.strzelecki@p.lodz.pl; 2Department of Biocybernetics and Biomedical Engineering, AGH University of Science and Technology, 30-059 Krakow, Poland; pioro@agh.edu.pl; 3Department of Diagnostic Imaging, Jagiellonian University Medical College, 31-501 Krakow, Poland

**Keywords:** texture analysis, matrix size, magnetic resonance, image quality

## Abstract

The selection of the matrix size is an important element of the magnetic resonance imaging (MRI) process, and has a significant impact on the acquired image quality. Signal to noise ratio, often used to assess MR image quality, has its limitations. Thus, for this purpose we propose a novel approach: the use of texture analysis as an index of the image quality that is sensitive for the change of matrix size. Image texture in biomedical images represents tissue and organ structures visualized via medical imaging modalities such as MRI. The correlation between texture parameters determined for the same tissues visualized in images acquired with different matrix sizes is analyzed to aid in the assessment of the selection of the optimal matrix size. T2-weighted coronal images of shoulders were acquired using five different matrix sizes while maintaining the same field of view; three regions of interest (bone, fat, and muscle) were considered. Lin’s correlation coefficients were calculated for all possible pairs of the 310-element texture feature vectors evaluated for each matrix. The obtained results are discussed considering the image noise and blurring effect visible in images acquired with smaller matrices. Taking these phenomena into account, recommendations for the selection of the matrix size used for the MRI imaging were proposed.

## 1. Introduction

One of the most important steps in the magnetic resonance (MR) image creation process is changing the size of the matrix. This process is steered by a radiographic technologist (MRI scanner operators) while considering the special requirements of an imaging specialist (radiologist). A radiologist is a physician not only involved in the clinical process, but also responsible for diagnostic image creation [1]. Diagnostic imaging plays an important role in clinical decision making, which involves establishing an accurate diagnosis (a judgment about changes in the morphology of internal organs and the tissues) as the initial mandatory step in a process of patient treatment [2].

The first step in medical imaging is setting the correct area for diagnostic imaging; therefore, it is necessary to select an appropriate anatomical structure that can be considered in the diagnostic process. This is represented by setting the field of view (FOV) of the body structure of the patient (Figure 1).

The image quality can be altered by setting the FOV. For example, setting a smaller FOV increases the image resolution. Such an image suggests the utilization of a higher gradient strength, which implies higher frequency-encoding gradients [3]. Such gradients produce an image of a given resolution (pixel size) and signal (pixel brightness). An increase in the frequency of encoding steps will force information from the K-space periphery and collect more spatial information than signals [4]. This is indicated by the K-space concept used for representing the Fourier-based transform in the time/frequency domain encoded using the complex sine and cosine wavelets produced by the scanner [5]. In this theoretical space, the periphery represents the spatial resolution, whereas the center represents the signal strength (Figure 2).

A slice can be selected if the appropriate anatomy is covered (an FOV is set). Slice thickness represents the amount of tissue included in the geometry of the voxel, and its value is responsible for the acquired image volume. Decisions concerning the voxel size are medical; therefore, some structures can be imaged every 4 mm (e.g., abdominal organs; large volume), whereas others (e.g., cranial nerves; smaller volume) need to be imaged in planes not exceeding a 1 mm width. The volume of the imaged object affects the signal: a greater volume implies a greater signal; an inversely smaller volume gives a weak signal. Thus, if a thinner slice is selected, a better resolution can be achieved but with a significantly weaker signal [6]. When the FOV size and slice thickness are set, the last decision of the radiographic technologist and the radiologist is the matrix size, which represents the image resolution. The resolution determines the ability to distinguish the smallest points as separate and adds “sharpness” to the image. The imaged structures will be finer with well-defined borders compared to imaged structures with well-defined tissue contrasts [7].

The term “tissue contrast” refers to the ability to differentiate between various tissues, and it is based on the intensity of the signal brightness. An increase in the matrix size suggests an increase in the frequency encoding readout, which implies an increase in the K space sampling in the columns (K-space is imaged as a plane rectangle or square) [8]. An increase in frequency encoding indicates the utilization of a higher frequency with a lower amplitude of the RF wave produced by the transmitter; this implies sampling from the K space periphery with more spatial information but few signals. An increase in the frequency readout domain and matrix size does not affect the scanning time because the process takes milliseconds during the free induction decay. However, setting the phase readout can affect the scanning time (proportionally) [9]. Besides the time penalty, which is secondary to an increase in the phase encoding steps, an increase in the phase readout will result in an isotropic image (square-shaped image matrix) that leads to a better image resolution. Furthermore, remember that the number of phase encoding steps never exceeds the frequency encoding steps owing to physical limitations [8].

Finally, the acquisition matrix may not be equal to the image matrix, whereas the second one can be reconstructed using an interpolation process. The objective of such interpolation is to obtain an image based on greater matrix size values and smaller pixels than those obtained from the acquisition process [10]. The most frequently used technique in the previously mentioned process is zero-filling interpolation, whereas zero-filled points (null information) are substituted with the signals of neighboring points that are effective in the K space periphery [11]. We did not use image interpolation in our study. A rule secondary to the K-space characteristic states that an increase in the gradient strength–matrix size and hence pixel size reduction decreases image quality as less signals will be withdrawn from the scanned object. Finally, any attempts to create a fine high-resolution image may result in a grainy image with a weak representation of different tissues anatomy [12]. The brightness of the pixel represents the signal strength, and it can be overcome by increasing the averages (consecutive scanning of tissue to build the K space signal); however, this incurs large time expenses that may not be accepted by the patient [13,14].

The resulting image, besides having a specific brightness, contrast, and resolution, is also often characterized by a certain texture [15]. This is a source of visual information about physical objects obtained by mapping their structures in an image. A texture is defined as a complex visual pattern that contains elements of a certain brightness, color, shape, etc. The properties of these elements correspond to the visual impression associated with a certain regularity, roughness, smoothness, granularity, directivity, etc., which characterize a given texture.

Texture is present in many biomedical images obtained using various medical imaging modalities that map the physical structure of tissues and organs [16,17]. The study of textural properties facilitates the determination and assessment of the characteristics of these organs; for example, the severity of lesions that occur in them. Furthermore, textural parameters are part of radiomics, wherein diagnostic information held in medical images is transformed into high-dimensional mineable data. Such information, obtained through objective and quantitative image analysis, is then used to build diagnostic support systems [18].

Texture analysis (TA) plays an important role in the assessment of MR images. Further, texture parameters (often along with other radiomic features) provide improved prognostic ability in advanced nasopharyngeal carcinoma [19]; predict renal function decline in patients with autosomal dominant polycystic kidney disease [20]; allow building models for accurate glioma grading by implementing information about perfusion, diffusion, and permeability from multimodal MRI data [21]; and provide an earlier prediction of response to chemotherapy and differentiated liver metastases from other liver tumors [22]. The significance of TA in medical imaging is illustrated by an increase in the number of publications in this area over the last decade, as indicated in Figure 3 (an even faster growth is observed for MR-related studies).

The repeatability and reproducibility of radiomic features in MR imaging (MRI), including the derived texture, have been the objectives of many studies; these studies were mostly performed on phantoms. Mayerhofer [23] demonstrated that texture features are highly sensitive to the signal to noise ratio (SNR) and spatial resolution, which results in different classification accuracies of T2w 3T MR images of polystyrene sphere phantoms. A complementary study demonstrated the significant effect of other MRI scanning parameters such as the number of experiments (NEX), flip angle, magnet strength, and scanner platform (GE vs. Siemens) on the distribution of the texture feature values estimated for MR gel phantom images [24]. A similar conclusion about the reproducibility of texture features calculated for fruit phantom images acquired using T1w, T2w, and FLAIR sequences was presented in [25], where only 15 (of 45) features represented certain repeatability in terms of concordance correlation coefficients. A higher feature reproducibility was observed for polystyrene phantom images but for different MR scanners (the obtained reproducibility was over 92% for the investigated 1.5T devices and ~80% for 3T scanners in terms of intraclass and concordance correlation coefficients) [26]. Of interest is that reproducibility of some texture parameters between two different scanners was also confirmed in [27], where predictive values of such features were investigated to discriminate benign and malignant thyroid incidentalomas in PET/CT imaging.

Furthermore, there are some studies on radiomic feature reproducibility performed on real MR data. The influence of image preprocessing (noise filtering, bias field correction) was analyzed on the stability of textural features extracted from multimodal glioblastoma MR images [28]. The investigated features were found to be sensitive to noise and artifacts, whereas the application of preprocessing algorithms improved their reproducibility. Another interesting study assessed the sensitivity of radiomics features to noise, resolution, and breast tumor volume in T1w and T2w MR images [29]. The analyses were performed for both the 2D and 3D ROIs; they found that most investigated texture features were sensitive to both noise and image resolution; however, the 3D features were more robust compared to the 2D features (single slice).

Based on these above-mentioned studies, it can be concluded that texture parameters are sensitive to image acquisition parameters, SNR, or the selection of MRI sequences. The reproducibility and repeatability of the texture features can be considered a measure of quality because these parameters affect image quality. To the best of our knowledge, influence of matrix size on image quality was studied based on phantom acquisition using the variable SNR value as a reference. However, important drawbacks of SNR as a parametric quality measure are known [15]. The SNR describes the amount of signal in relation to the background noise but does not reflect changes and the preservation of a signal in the image parts which represents anatomical structures. Such changes are the most important for the radiologists. These limitations can be omitted with the use of a proposed approach of textural features correlation. Thus, in this study a novel approach is proposed, the reproducibility of texture parameters is used to evaluate the effect of one of the important parameters of image acquisition (the size of the imaging matrix) on image quality. Matrix size has a significant impact on image quality, which implies the necessity of the very accurate estimation of quality changes. The contribution of this paper is demonstrating that TA (understood as the study of the correlation of texture parameters determined for the same tissues visualized in images acquired with different matrix sizes) enables the selection of the optimal matrix size. Such a matrix can help ensure good image quality, which is considered the best representation of tissue structure properties described by texture parameters.

The aim of the study was to evaluate the effect of the matrix size on the preservation of image quality assessed with the help of texture features values. The analysis was performed for separate tissues that represent different textures in the MR image. Another aim of the work was to show how changing the size of the matrix affects the radiological image, which can help in creating guidelines for modifying the scanner parameters with the greatest safety margin depending on the image quality.

## 2. Materials and Methods

The study protocol was designed based on guidelines of the Declaration of Helsinki and the Good Clinical Practice Declaration Statement. All images were anonymized before processing to ensure personal data safety. Furthermore, written acceptance was obtained from the Local Ethics Committee to conduct this study (No. 155/KBL/OIL/2017 dated: 22 September, 2017). Data from 20 patients (12 women and 8 men) between the ages of 34 and 61 years were used in the study. T2-weighted coronal images of shoulders were obtained during normal diagnostic procedures with added sequences with changed matrix size. The exclusion criteria included patients with implants or other foreign bodies that produced susceptibility artifacts; furthermore, movement during the examination was considered a negative selection criterion because it caused image artifacts that affect the image analysis. To provide stable conditions for the image creation, patients with BMI in the range of 20–26 were included in the study. The T2-weighted coronal sequences of the shoulder were analyzed (Figure 4). For this study, we selected images of tissues that were free of pathologies. Images were selected to satisfy stable (repetitive) conditions for texture feature analysis.

MR data were acquired from a 1.5 system Siemens Essenza (Erlangen, Germany) equipped with 12 dedicated table coils and eight channel shoulder coils using a gradient strength of 30 mT/m and a slew rate of 100 T/m/s. For the shoulder coronal images, the following parameters were selected: applied echo time = 102 ms, repetition time = 3200 ms, flip angle = 150°, phase oversampling = 100, and distance factor = 20. The scan geometry was changed from 256 × 256 to 320 × 320, 384 × 384, and 448 × 448 up to 512 × 512. Slice thickness = 3 mm was used with an FOV read = 200 mm and FOV phase = 100. Those parameters were used as a standard to create a T2 weighted pulse sequence although different matrices which were used to test the influence of the matrix size parameter on the preservation of tissue features in the created image. A voxel of one plane non-isotropic resolution at 0.8 × 0.8 × 3 mm was acquired for matrix size 256 × 256. For other matrices, voxel sizes along with other parameters are shown in Table 1. All other parameters were kept constant by scanning at different matrix sizes. However, scanning with a matrix size of 512 × 512 induced a decrease in TE to 98 ms with an increase in the echo spacing from 2.3 ms to 2.8 ms, which finally resulted in a decrease in echo train per slice from 60 to 55 (5.5%), which was caused by an increase in the number of phase encoding steps (while scanning was performed on a higher matrix with other geometrical parameters kept constant). This resulted in a slight decrease in scanning time from 3.2 min to 2.8 min. The above changes may slightly influence the image quality; however, this was omitted as a concern for the extreme values of the matrix. An increase in the bandwidth value for the 512 × 512 matrix from the 180 Hz/Px to 181 Hz/Px was negligible.

After storage and anonymization, the images were postprocessed with dedicated qMaZda software (developed at the Lodz University of Technology, Institute of Electronics, Lodz, Poland [30]).

The ROI was visualized and analyzed for every patient on five consecutive layers of each sequence (Figure 5). The average ROI sizes depend on the acquisition matrices; they are presented in Table 2. The position of the ROI in the image was determined by the radiologist. The ROI size was calculated in proportion to the FOV and, consequently, to the image area and size of each tissue tested. The stability of the position and size of the ROI in the image was monitored in all examined cases.

Such ROI sizes are sufficiently large to represent statistical textural features. Analyses were performed for the 2D case, and texture parameters were averaged for ROIs defined on five selected layers and the 3D case, wherein three-dimensional texture features were calculated for the volume build from five subsequent ROIs.

The results of texture analysis depend on the image acquisition parameters. Different values of such parameters (like e.g., variable matrix size used in this study) may cause the brightness and contrast variation in individual regions of interest. As a result, the values of some texture features depend not only on the texture, but also on the ROIs brightness and contrast. For this reason, some of the features describe not only the structure of the tissue under examination but also the scanner’s uneven sensitivity within the analyzed tissue region. This may in turn lead to an inappropriate description of the tissue. To limit these phenomena, the ROIs are normalized. Normalization is the stretching of the histogram in the ROI into the entire available intensity range. This improves the contrast of the investigated texture and reduces the influence of the ROI local mean intensity. Both effects improve the quality of the features obtained, which can lead to better classification results of MR data, as indicated in [31]. Thus, analyses were performed both for raw and normalized ROIs.

The normalization procedure for 8-bit images is described by:(1)$$I_{norm}\left(x,y\right)=\{\begin{array}{ccc} 255 & for & N\left(x,y\right)>255\\ N\left(x,y\right) & for & 0\leq N\left(x,y\right)\leq 255\\ 0 & for & 0<N\left(x,y\right) \end{array}\;$$
where
$$N\left(x,y\right)=\;\mathrm{round}\_\mathrm{to}\_\mathrm{int}\left(255*\frac{I\left(x,y\right)-min_{norm}}{max_{norm}-min_{norm}}\right)$$

*min_norm_*, *max_norm_* represent the minimum and maximum normalized values, respectively; and *N*(*x*,*y*) and *I_norm_* (*x*,*y*) represent the original (raw) and normalized images, respectively.

The ±3σ normalization method was used in this study. The range of intensities is defined as *min_norm_ = μ − 3σ* and *max_norm_ = μ + 3σ*, where *μ* represents the mean intensity and *σ* denotes the standard deviation of the image intensities in the ROI. As a result, all original ROI intensities are mapped to this new intensity range, improving the ROI contrast. Such normalization is useful when the intensity histogram of the texture is close to a Gaussian distribution.

The following 310 texture features were calculated for each analysis:

11 gray-level co-occurrence matrix (GLCM) parameters [32] calculated in four directions: vertical, horizontal, 45°, and 135°. Five distances between the pixels (1–5) were considered. In total, 11 × 4 × 5 = 220 parameters were calculated.Five run-length matrix (RLM) parameters [32] for the co-occurrence matrices; the calculations were performed in four directions, yielding 20 parameters.Five gradient matrix parameters for the image gradient matrix initially built with a high-pass filter using a 3 × 3 mask [33].Five parameters defined for the first-order autoregressive model (AR), which assumed that the brightness of a given image pixel depends on the weighted sum of the neighboring pixels [34].Sixteen parameters calculated based on the Haar wavelet transform (HW). The data represent the energy of the ROI sub-images after conversion to the wavelet coefficient space. The wavelet transform was calculated for four image scales, which resulted in four downscaled sub-images for each scale, which yielded sixteen parameters in total [35].Nine first-order features (HIST) and histogram-based features (mean, variance, skewness, kurtosis, and percentiles: 1,10, 50, 90, and 99%, respectively).Magnitudes of Gabor transform (GAB) calculated in four directions: vertical, horizontal, 45°, and 135°. Six sizes (4, 6, 8, 12,16, 24) of the Gaussian envelope were considered. In total, 4 × 6 = 24 parameters were calculated.Eight bins of histogram of oriented gradients (HOG) that counts occurrences of gradient orientations.

The feature values of the texture parameters were calculated for each patient and acquisition matrix; the results were obtained for all patients and correlated in pairs for any matrix. Following the methodology described by [15,17], Lin’s concordance correlation coefficient (*ρ_c_*) was estimated between the textural features obtained for different acquisition matrices. In each analysis, the sets of textural features (obtained for all patients, for which the given structures were visualized) were considered. Equation (2) defines *ρ_c_* as:(2)$$\rho_{c}=\frac{2\rho \sigma_{i}\sigma_{j}}{\sigma_{i}^{2}+\sigma_{j}^{2}+{\left(i-\mu_{j}\right)}^{2}}$$
where *μ_i_* and *σ*^2^*_j_* represent the means and standard deviations of the texture feature vectors, respectively; *(i*,*j)*
*∈* [1,…,5], *i ≠ j* correspond to the five analyzed matrices; and *ρ*_c_ represents the correlation coefficient between these two vectors.

## 3. Results

### 3.1. Dependency of the Texture Parameters Correlation with Matrix Size Transitions

Plots presented in Figure 6 show the average correlation (estimated for all texture features) with transitions from matrices one step up according to their size. Each tissue behaves slightly differently; however, tissues representing bone and fat show a higher correlation for all transitions when compared to correlations obtained for muscle. Thus, texture information for tissues with more complex structures is more preserved when compared to that for muscle. Furthermore, a greater restriction of texture information occurs because it is represented by a smoother region with limited and directional intensity variations. An interesting effect of normalization can be observed: changes in the correlation value between the various tissues decrease; for the muscle, they become similar for the remaining two more “structural” tissues. For matrices 448 × 448, 384 × 384, and 320 × 320, the averaged correlation is at an acceptable level of 0.65–0.8 for all tissues. Although the normalization reduces the maximum correlation (mostly for fat), the range of matrices for which the transition is described by a high correlation is 448 × 448 and possibly 384 × 384.

The 3D analyses results indicate similar tissue properties: textures of fat and bone are more resistant than muscle texture to matrix changes in terms of higher average correlation coefficient values. Furthermore, a higher value of this coefficient is observed for bone tissue when compared to that in the 2D analysis. Image normalization reduces differences between the correlation coefficient values estimated for all analyzed tissues, and it increases the range of matrices for which this coefficient has a similar value. Further, for the muscle, normalization flattens the variability in the correlation coefficient at transitions between all matrices.

### 3.2. Detailed Analysis of Different Texture Parameter Types

The highest values of Lin’s correlation coefficient are observed for features derived from the gray-level co-occurrence matrix (GLCM) for the 2D and 3D cases (Figure 7). In addition, the HOG and DWT parameters preserve a high correlation for the majority of cases. This is also observed for muscles with large matrices, wherein the autoregression model (ARM) and GLRLM features are good ones. The highest correlation, especially for highly structured textures, is noted for the first-order (histogram) features; this is understandable because the change in matrix size does not modify the MR signal intensity for all visualized tissues. A structured texture is one represented by clearly defined primitives (microtexture) along with placement rules defining their spatial distribution. Such a distribution creates a specific arrangement of primitives, which builds a macrotexture [15]. Furthermore, this texture can appear in the MRI of tissues characterized by strong signals, which allows the detailed visualization of the tissue structure.

In this study, structured textures represent fat and bone tissues; in the latter, bone trabeculae constitute microtexture primitives, whereas their distribution in cancellous bone defines a macrotexture. The opposite is the case with the texture of the muscle, wherein the structure of this tissue is hardly visible because of the small MR signal. For such a texture, it is difficult to observe primitives; they are represented by a smooth gray-level distribution.

### 3.3. Acquisition Times

Acquisition times are recorded for all analyzed patients, as shown in Figure 8. For most matrices, the time increases linearly with an increasing matrix size. The scan increases proportionally to a larger size because an increase in both matrix sizes indicates an increase in the number of phase encoding steps. For the 512 × 512 matrix, a reduction in acquisition time is observed for most patients. This is caused by an increase in echo spacing and the secondary reduction of echo train per slice as an effect of the forced increase in the phase encoding steps.

### 3.4. Investigation of the Normalization Effects

The calculations of texture parameters correlation coefficients were repeated also for normalized ROI. The ±3σ normalization method was applied, as described in Section 2.

Figure 9 and Figure 10 show plots that demonstrate normalization’s influence on the averaged values of correlation coefficients. There is a need to focus on the low average changes in the correlation coefficient values with transitions between all pairs of matrices using normalization (the effect is visible for the 2D analysis). A 3D analysis slightly increases the correlation coefficients for all types of tissues.

## 4. Discussion

There are many factors that affect the MR image quality in addition to their texture information. We focused on two factors: SNR and image blurring. A smaller matrix for the same FOV leads to a smoother image (the signal in larger voxels is more averaged), and therefore, the reduced image resolution introduces some amount of blurring. Both these phenomena are quantified as illustrated in Figure 9 and Figure 10. We considered the average amplitude of the image gradient estimated for each ROI to evaluate blurring. The gradient amplitude decreased in the blurred images. Thus, we evaluate image sharpness (the inverse of blurring) as the value of the gradient amplitude to ensure that the degree of blurring is consistent with the change in SNR. Since a higher SNR indicates better image quality, the same occurs with image sharpness; a larger value reflects better image quality.

Based on Figure 11 and Figure 12 it can be observed that the noise increases with an increase in the matrix size, and therefore, the SNR values suggest selecting smaller matrices (256 × 256 or 320 × 320) for bone and fat. On the other hand, the sharpness increases with an increase in the size of the matrix, which also affects image quality. However, the measure of image sharpness depends on the amount of noise in the image, and thus, its highest value does not necessarily represent the best image quality (more noise may lead to larger gradients in the image). Figure 10 shows that, for all tissues, matrices of 384 × 384 or larger afford an increasing image sharpness.

A significant blur can be observed for 256 × 256 and 320 × 320 matrices in the case of bone and fat; this causes the loss of “textural” information. A similar effect occurs for the 512 × 512 matrix, and it is attributed to the higher noise. Thus, from the point of view of preserving the values of the texture parameters (and thus information about the tissue structure), the 384 × 384 and 448 × 448 matrices are the best. This is confirmed by the correlation values for transitions between these matrices (Figure 6, especially when normalization was applied); this is in line with results presented in ref. [22], wherein a larger number of reproducible texture features are observed for higher-resolution images for T2w phantom images. The values of averaged correlation coefficients are similar for both tissues, and this results from the fact that bone and fat have a complex structure, which translates into a texture with similar properties such as roughness, granularity, and coarseness.

Slightly different conclusions can be drawn for the images of muscles, and this is considerably smoother and has a lower contrast compared to other tissues; this is shown quantitatively by the sharpness measure, which is clearly lower than that for bone and fat (Figure 12). There is more noise in the muscle image than in the case of other tissues because of the smaller signal. Such lack of clearly visible muscle structure and high noise reduces correlation coefficients during between-matrix transitions. However, a larger matrix preserves more details and reveals the structure of this tissue, which is blurred for small matrices. This causes the texture to become more complex, and therefore the values of correlation coefficients increase for larger matrices (Figure 6); both for 2D and 3D analyses (especially for no normalized data). Consequently, considering the SNR, sharpness measures and the regeneration of texture properties for larger matrices, the 384 × 384 and 448 × 448 matrices again seem to be the best choices for muscle tissue.

Normalization minimizes the effect of matrix size on the loss of texture information; however, there is a trade off in terms of a slight global reduction in the correlation coefficient values. Normalization should be recommended when, for some reason, image acquisition is performed for extreme matrices, or if imaging is performed for several sizes of matrices.

For the 2D analysis, the low average changes in the correlation coefficient values with transitions between all pairs of matrices were observed after ROI normalization. A 3D analysis slightly increases the correlation coefficients for all types of tissues, and therefore, if there is such a possibility, the analysis should be conducted because the resulting 3D texture can more accurately reflect the structure of the tissue. This conclusion appears in many research papers; for example, Ortiz-Ramon [35] demonstrated that 3D MRI texture features ensure a better classification of brain metastases from lung cancer and melanoma; Xu [36] reported the better predictive performance of 3D texture features across three cancer types (intrahepatic cholangiocarcinoma, high-grade osteosarcomas, and pancreatic neuroendocrine tumors) with multimodality (CT and MRI) imaging. Fetit [37] outlined that 3D textural features extracted from T1w and T2w images can improve the diagnostic classification of childhood brain tumors when compared to their 2D counterparts.

Furthermore, our findings confirm the results presented in a previous study [26], wherein the texture features estimated in the 3D analysis are less sensitive to noise and the change in the resolution when compared to 2D features (analysis of MR breast cancer tumor images). Phantom studies performed on 1.5T and 3T scanners indicated that the 3D features did not improve repeatability (compared to 2D parameters) [23]. Further, an additional analysis along the Z-plane can introduce feature instability after phantom repositioning (acquisition was performed twice for different phantom locations).

The highest correlation coefficient values were observed for GLCM-derived texture features. This is in line with many reported studies on MR image analysis, wherein such parameters often provide the best classification or segmentation results. In addition, GLCM parameters ensured the best classification results regardless of the changes in the MR acquisition protocol parameters, which is demonstrated in a phantom study [20]. A similar conclusion was derived for DWT parameters because the wavelet transform is a very efficient tool for texture image analysis. For tissues such as muscle, which represents high directionality, GLRM-derived features also perform well especially when visualized with lower-size matrices; they can efficiently describe the directional properties of such textures.

Examinations were performed only for bone, fat, and muscle tissues, which is a limitation of our study. Although images of such tissues show large textural variety, they represent extreme cases in this respect among all tissues of the human body: from “smooth” fat, “smooth and stripped” muscle, and “trabecular” bone. Therefore, it was assumed that the analyzed images covered most of the possible pattern variability represented by human tissues. The obtained Lin’s correlation values demonstrated that images acquired for larger matrices indicate the most texture information. Thus, the 384 × 384 and 448 × 448 matrix sizes are the best choices for most tissues. These matrices can provide the best quality images because the texture represents a visualized tissue structure; this is in line with the medical experience from daily practice, and it shows a usability of the 320 × 320 and higher matrix sizes (possibly rectangular if the FOV phase is not 100%). There is a need to increase the matrix size to provide the best quality images; this is clearly demonstrated in our study. A further increase in the matrix size up to the 512 phase encoding steps provides no further benefits because the signal is lost.

Another limitation is the relatively small group of patients. The analysis of the image textures obtained for 20 patients allowed us to draw statistically reliable conclusions. The concept of using texture features preservation worked well with much coarser images, such as ultrasound [38], collected from relatively small groups of patients In this study, we used a 1.5T scanner, which is perfectly adequate for most clinical applications; however, it would be an interesting repetition of the examinations for 3T MRI, which allows the analysis of texture parameters for more accurately imaged tissues by providing images with higher SNR. Indeed, research was performed only for the T2-weighted sequence. It is certainly worth repeating it for other “anatomical” sequences like T1 and “fluid signal sensitive” sequences like Short-TI Inversion Recovery (STIR).

## 5. Conclusions

It is extremely important for medical professionals to understand the effect of matrix size on textural classification for imaging different tissues. The selection of matrix size is a crucial step in MR scan planning. The awareness of the radiologist and radiographic technologist with regard to the true effect of the matrix selection on image quality is another step when selecting the best possible options to ensure the highest image quality in a reasonable scanning time. In this study, we quantitatively showed how the amount of texture information in visualized tissue is influenced by changes in the matrix size. We found the following:

Matrices with sizes 384 × 384 and 448 × 448 are the best choice for all types of tissues because the values of texture parameters are mostly preserved; thus, information about the tissue structure can be maintained.Image normalization is recommended when image acquisition is performed for large matrices because it reduces the effect of matrix size on the loss of texture information; however, a slight global reduction of correlation coefficient values for texture parameters was observed. 3D image texture reproduces visualized tissue more accurately than the texture of 2D cross-sections; therefore, a 3D approach is recommended if a quantitative image analysis needs to be performed.Knowledge regarding the properties of the image acquired with different matrix sizes supports radiologists and radiographic technologists during MR image acquisition planning when obtaining the best quality of medical images within an acceptable time.

## Figures and Tables

**Figure 1 jcm-11-02526-f001:**
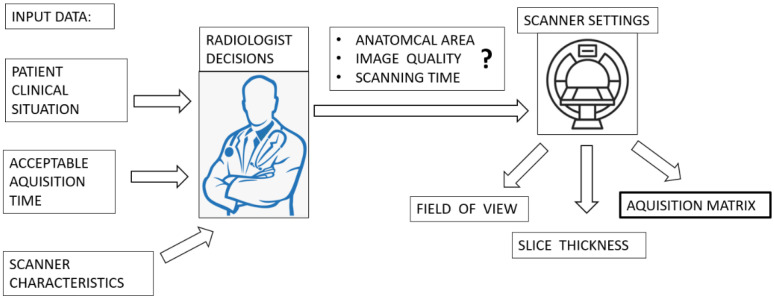
Block diagram presents MR scanner position in the workflow algorithm of radiological decisions made before scanning patients. Notice importance of matrix size setting in the decision process.

**Figure 2 jcm-11-02526-f002:**
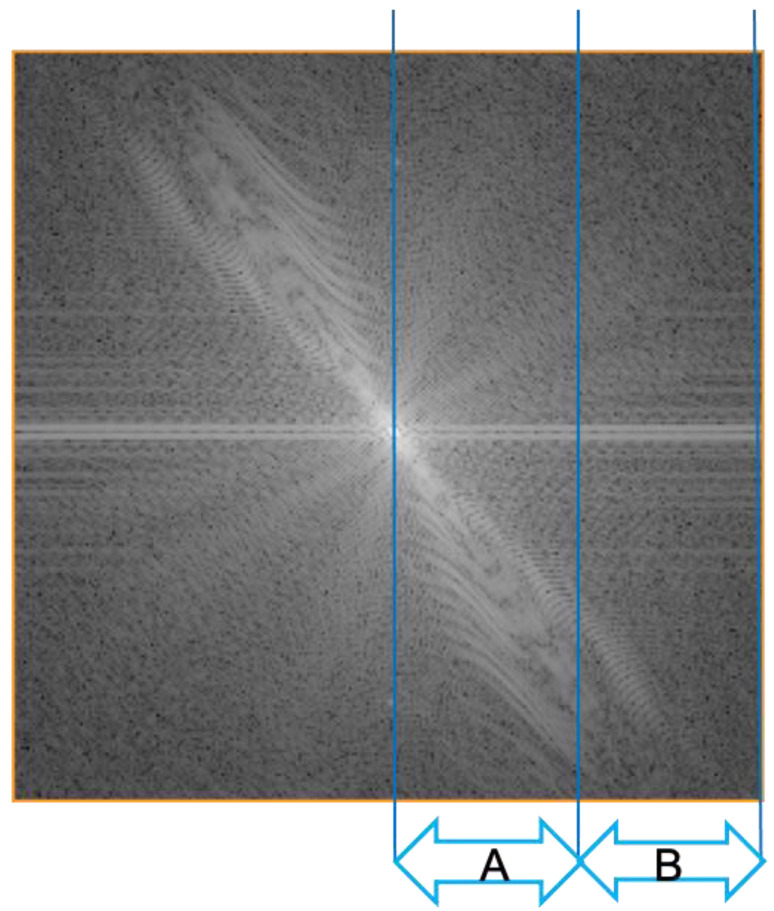
Graphic representation of K-space concept. A represents the center of the K-space area where the signal strength is encoded. B indicates the K-space periphery and presents the encoded spatial information of the image. A gradient frequency modulates the alignment of the signal, and it is set by the MR machine operator when the image acquisition matrix is selected.

**Figure 3 jcm-11-02526-f003:**
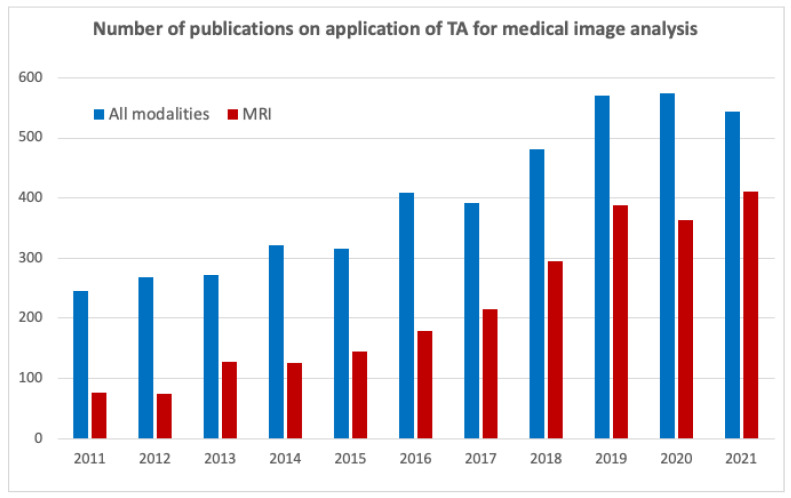
Number of publications on TA of medical images according to the Scopus database (accessed on 01 February 2022, www.scopus.com).

**Figure 4 jcm-11-02526-f004:**
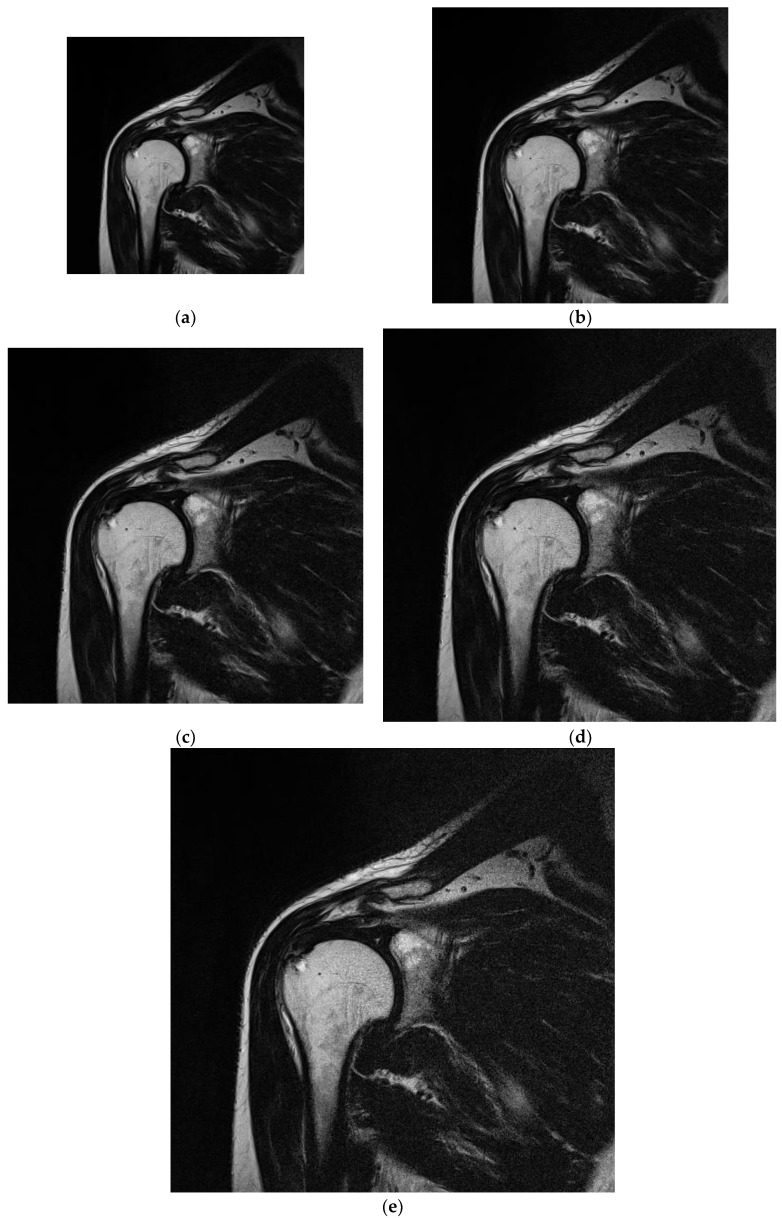
Sample analyzed images acquired for matrices with various sizes under the same FOV. (**a**) 256 × 256; (**b**) 320 × 320; (**c**) 384 × 384; (**d**) 448 × 448; (**e**) 512 × 512.

**Figure 5 jcm-11-02526-f005:**
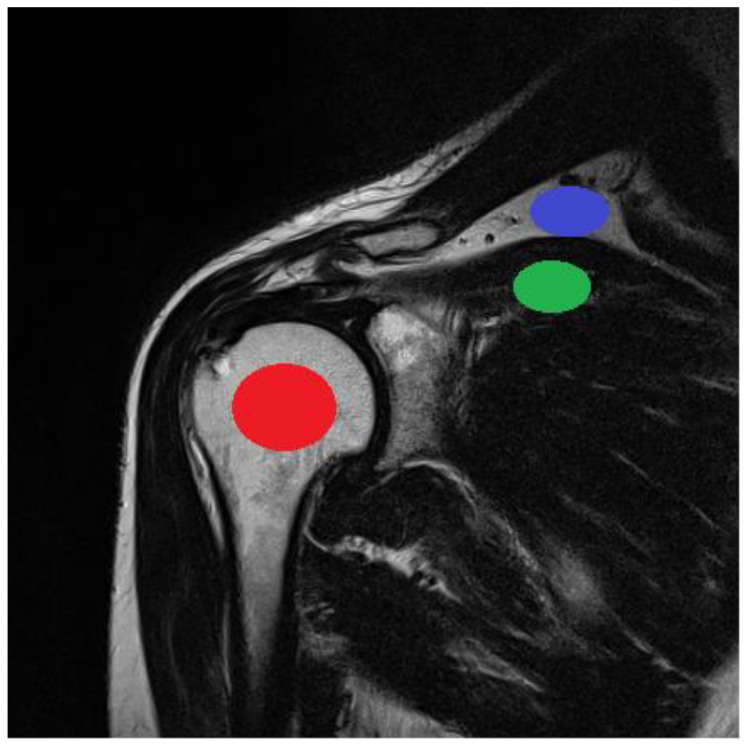
Position of ROIs used for analyzing different tissue types (bone: red; fat: blue; and muscle: green).

**Figure 6 jcm-11-02526-f006:**
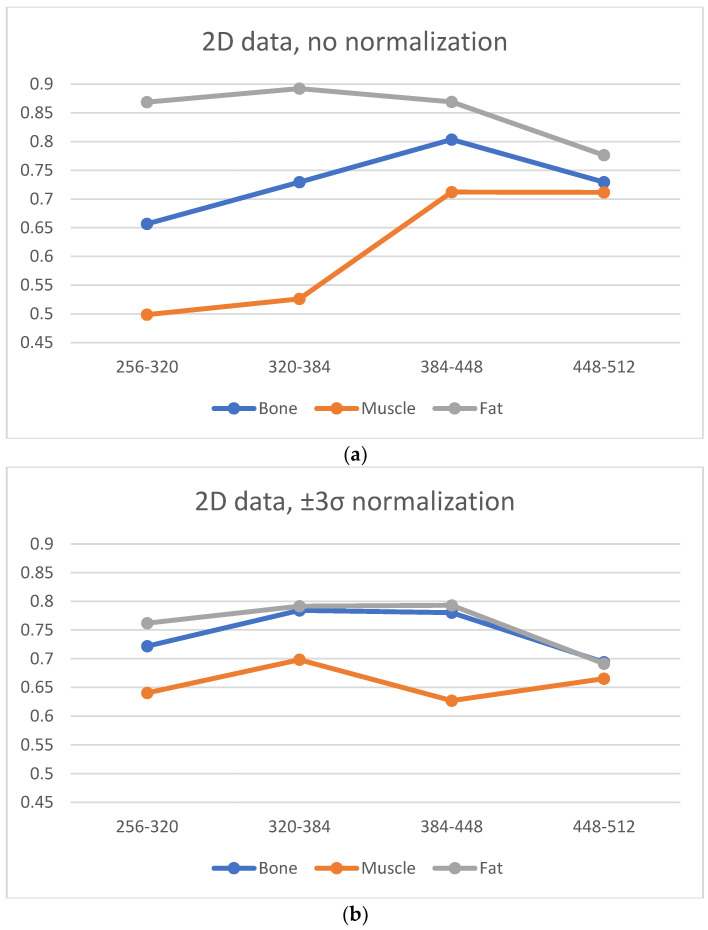
Average Lin’s coefficients for all texture features evaluated between neighboring matrices.

**Figure 7 jcm-11-02526-f007:**
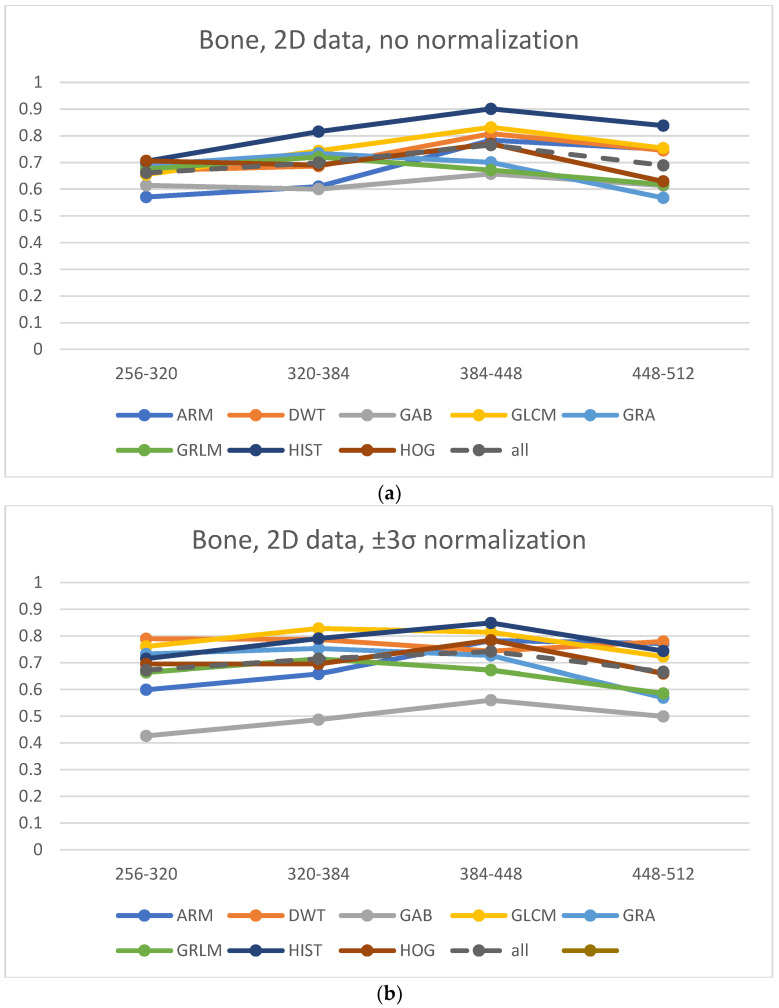
Average Lin’s coefficients for analyzed texture features evaluated between neighboring matrices. ARM—Autoregressive Model, DWT—Discrete Wavelet Transform (Haar wavelet), GAB—Gabor Transform, GLCM—Grey Level Coocurrence Matrix, GRA—Gradient matrix, GRLM—Grey Level Run-Length Matrix, HIST—Histogram, HOG—Histogram of Oriented Gradients.

**Figure 8 jcm-11-02526-f008:**
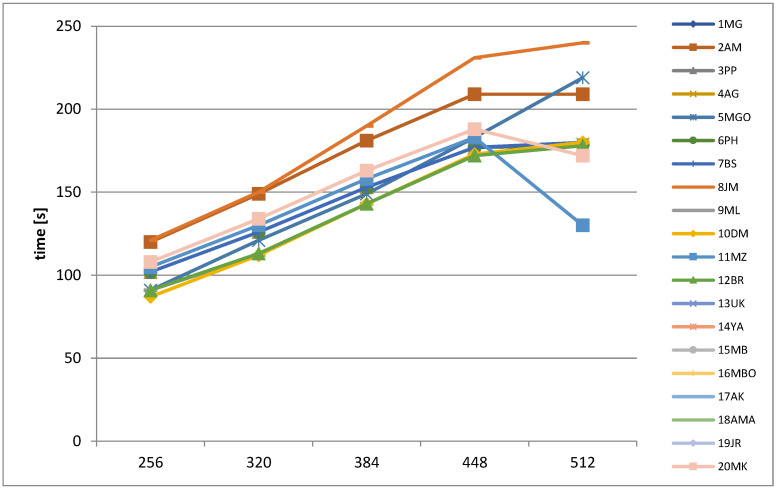
Acquisition times for all patients.

**Figure 9 jcm-11-02526-f009:**
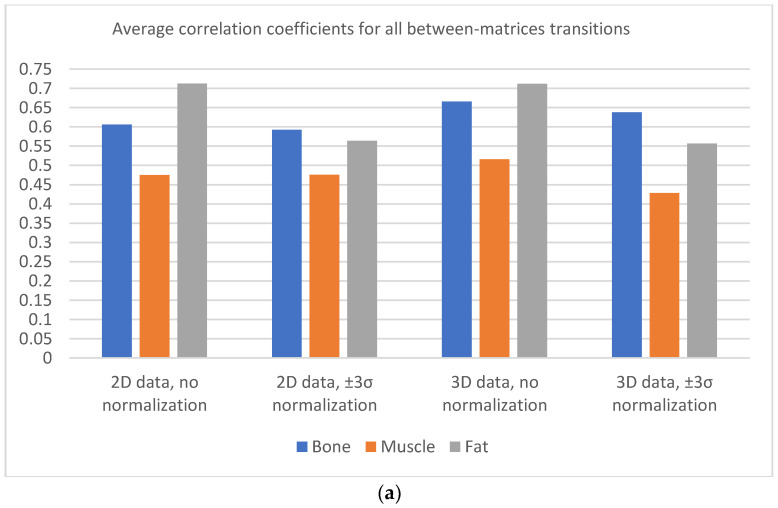
(**a**) Values of correlation coefficients averaged at all transitions between matrices; (**b**) mean change of correlation coefficients at all transitions between the matrices.

**Figure 10 jcm-11-02526-f010:**
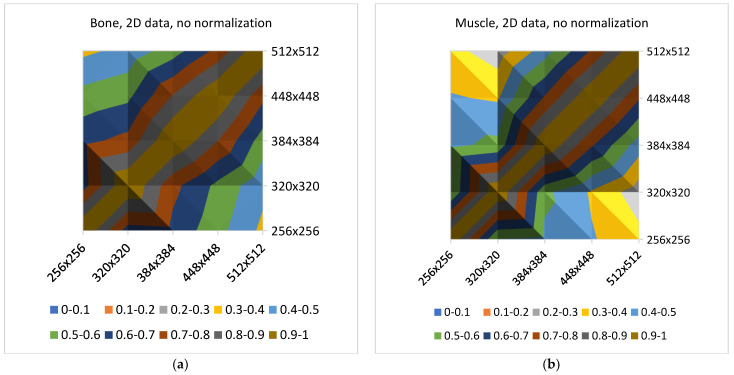
Effects of normalization on the values of correlation at all between–matrices transitions. No normalization: (**a**) bone, (**b**) muscle; Normalization: (**c**) bone and (**d**) muscle.

**Figure 11 jcm-11-02526-f011:**
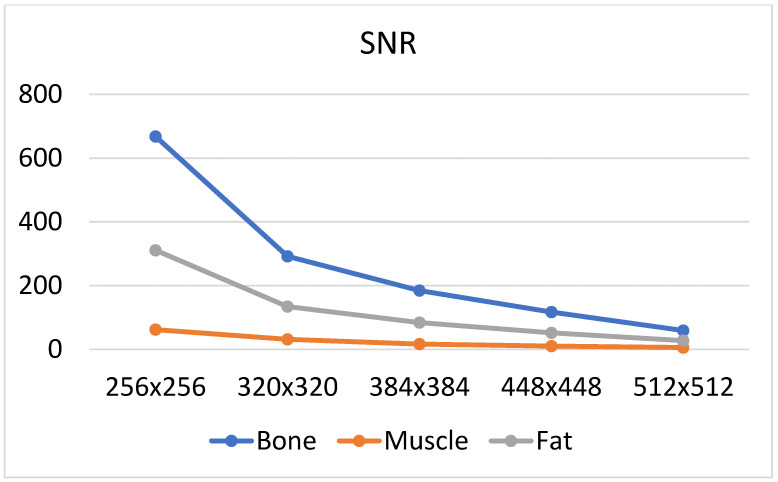
Mean SNR values (averaged for all patients) evaluated for all matrices and analyzed tissues.

**Figure 12 jcm-11-02526-f012:**
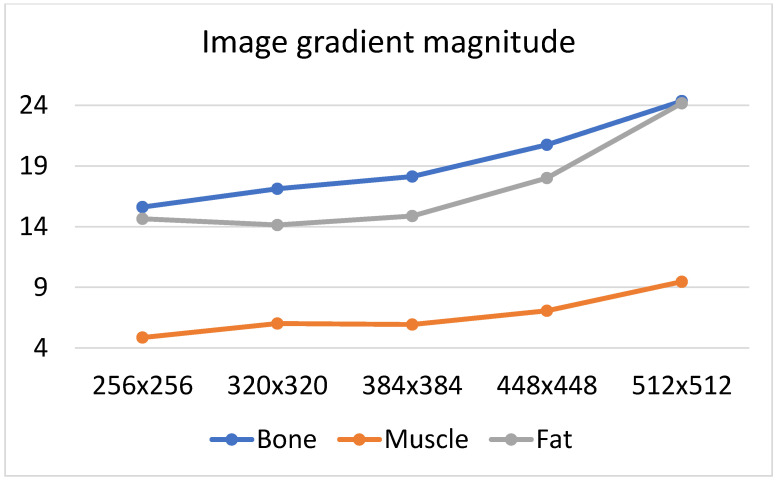
Image sharpness evaluated for all matrices and analyzed tissues.

**Table 1 jcm-11-02526-t001:** Scanning parameters according to different matrix size. TE—time echo, TR—repetition time, FA—flip angle, PA—phase oversampling, DF—distance factor, FOV—field of view, VOX—voxel size, AV—averages.

	256 × 265	320 × 320	384 × 384	448 × 448	512 × 512
TE	102	102	102	98	98
TR	3200	3200	3200	3650	3650
FA	150	150	150	150	150
PA	100	100	100	100	100
DF	20	20	20	20	20
FOV	200	200	200	200	200
VOX	0.8 × 0.8 × 3	0.6 × 0.6 × 3	0.5 × 0.5 × 3	0.4 × 0.4 × 3	0.4 × 0.4 × 3
AV	2	2	2	2	2

**Table 2 jcm-11-02526-t002:** Average ROI sizes for various matrices [pixels].

Matrix Size	256 × 265	320 × 320	384 × 384	448 × 448	512 × 512
Bone	1165	1723	2348	3167	4185
Muscle	652	934	1291	1711	2241
Fat	471	668	893	1224	1590

## Data Availability

The data presented in this study are available on request from the corresponding author. The data are not publicly available due to property rights of the providing institution.

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
