# Peer review of "Effect of Matrix Size Reduction on Textural Information in Clinical Magnetic Resonance Imaging"

_jcm, 2022, doi:10.3390/jcm11092526_

Round 1
Reviewer 1 Report
Peer Review – manus no 1642576
Abstract.
Please don’t use abbreviations with presenting without presenting their meaning e.g. MRI, SNR.
Introduction
Overall, very well written.
The introduction is very long.
The word technologist – normally the European term would be radiographer or a radiographic technologist. Consider rephrasing.
Consider rephrasing. It is not correct that a radiologist is responsible for the image creation – that would be the radiographer or technologist. But the radiologist is responsible for diagnosing diagnosis on the basis of X-rays imaging or scans. The radiologist choice eg T2 or T1, slice thickness. Of course, this can be different from country to country.
Typically, a radiologist will assess whether images have the diagnostic quality, within the scan protocol accepted by a group of radiologists. But if the scan time is very long, the radiographer must make some choices about image acquisition/FOV/k-space and more. Therefore, the scan parameter and setup are radiographers’ area of expertise.
Interesting figures.
Figure 2. MR machine operator? Do you mean a radiographer or radiographer technologist?
Consider subheadings. This will create a better overview of the paper.
Consider adding a purpose to the introduction
Method section
Written acceptance – is that the same as written informed consent? What about oral consent?
How many patients was excluded – and why?
Please consider to add all the scan parameter into a table. Gives a better overview and understanding of the scan parametres.
Why shoulder region? Any thought about this? Is shoulder a common MRI investigation?
Consider - Can body size play a role in this?
Consider - How can body size affect the results?
What about diff between males and females? Any thoughts?
Many figures – are all necessary? Consider to place figures side by side, to create a overview and comparison.
Overall, what about the scan time – the scan time will always be affected when we adjust the scan parameters.
Discussion.
Why add phantom results/figure?
Did you scan patients and a phantom? The methods section does not include any phantom?
Conclusion
Ok.
Overall interesting topic and paper.
Author Response
We would like to thank the Reviewer for valuable comments, that significantly improved the quality of this paper. All comments, if applicable, were addressed in the paper text.
Abstract.
Please don’t use abbreviations with presenting without presenting their meaning e.g. MRI, SNR.
Suggested corrections in the manuscript text were introduced.
Introduction
Overall, very well written.
Thank you for this comment that is important to us.
The introduction is very long.
We considered reducing the introduction again. However, we believe that all its content is important as it brings the reader closer to the topic. The introduction is a bit extensive because, among others, describes the role of selected parameters in the MR image creation process. We believe this may be of value to readers unrelated to MR image acquisition technology but interested in MR image analysis. It can also be helpful for medical specialists unfamiliar with the field of imaging diagnostics to gain a better understanding of the processes behind decision making when selecting MR imaging protocols.
The word technologist – normally the European term would be radiographer or a radiographic technologist. Consider rephrasing.
Thank you for this observation. term “radiographic technologist” is far more accurate than “technologist” according to the existing nomenclature, this has been changed in the text.
Consider rephrasing. It is not correct that a radiologist is responsible for the image creation – that would be the radiographer or technologist. But the radiologist is responsible for diagnosing diagnosis on the basis of X-rays imaging or scans. The radiologist choice eg T2 or T1, slice thickness. Of course, this can be different from country to country.
Thank you for this remark. Indeed there was mental acronym what is not appropriate here. Radiographic technologist is responsible for the image creation as he/she has to made a proper choice from the plethora of the settings. However, this process is under supervision of the radiologist who controls the effect of radiographers action and sometimes, if needed, he guides the radiographers. Those relations may vary between different radiological departments and depends mostly of knowledge of the doctor and the technician who are involved in the creation of the specific MR image.
Typically, a radiologist will assess whether images have the diagnostic quality, within the scan protocol accepted by a group of radiologists. But if the scan time is very long, the radiographer must make some choices about image acquisition/FOV/k-space and more. Therefore, the scan parameter and setup are radiographers’ area of expertise.
We agree with this observation. There is tendency among radiologists to improve the quality of the scans what results in the increase of the scanning time. Radiographic technologists uses their knowledge and experience to manage best possible images in the reasonable scanning time.
Interesting figures.
Thank you for this kind remark.
Figure 2. MR machine operator? Do you mean a radiographer or radiographer technologist?
Thank you for this remark – radiographic technologist will be more appropriate term describing the role of this person, who is responsible for the technical settings on the scanner. Appropriate changes were made in the text of the manuscript.
Consider subheadings. This will create a better overview of the paper.
Subheadings were added to the section Results; now it should be much easier to follow.
Consider adding a purpose to the introduction.
Thank you for this remark. The appropriate paragraph was added to the section “introduction”:
The aim of the study was to evaluate the effect of the matrix size on the preservation of image quality assessed with the help of texture features. The analysis was performed for separate tissues that represent different textures in the MR image. The possible influence of image processing techniques on the preservation of the value of tissue texture features was analyzed. Another aim of the work was to show how changing the size of the matrix affects the radiological image, which can help in creating guidelines for modifying the scanner parameters with the greatest safety margin depending on the image quality.
Method section
Written acceptance – is that the same as written informed consent? What about oral consent?
The Bioethics Committee of the Jagiellonian University gives its written consent to conduct the study and requires the written consent of the experiment participants in the case of a medical experiment. It is not required for retrospective studies.
How many patients was excluded – and why?
14 patients were excluded from the study because their images did not conform criteria of the study as: presence of susceptibility artifacts , movement (even of small amplitude like “shaking”) what produces unclarities to the image. As well as the presence of other artifacts what may interfere with image analysis. Patients with BMI 20- 26 were included to the study what means that selection included body mass.
We added appropriate paragraph describing the exclusion criteria in the Method section.
Please consider to add all the scan parameter into a table. Gives a better overview and understanding of the scan parametres.
The table with parameters was added as requested.
|
256 × 256 |
320 × 320 |
384 × 384 |
448 × 448 |
512 × 512 |
TE |
102 |
102 |
102 |
98 |
98 |
TR |
3200 |
3200 |
3200 |
3650 |
3650 |
FA |
150 |
150 |
150 |
150 |
150 |
PA |
100 |
100 |
100 |
100 |
100 |
DF |
20 |
20 |
20 |
20 |
20 |
FOV |
200 |
200 |
200 |
200 |
200 |
VOX |
0,8x0,8x3 |
0,6x0,6x3 |
0,5x0,5x3 |
0,4x0,4x3 |
0,4x0,4x3 |
AV |
2 |
2 |
2 |
2 |
2 |
Table Scanning parameters according to different matrix size. TE - time echo, TR - repetition time, FA - flip angle, PA - phase oversampling, DF - distance factor , FOV - field of view, VOX - voxel size, AV – averages.
Why shoulder region? Any thought about this? Is shoulder a common MRI investigation?
Shoulder was selected because it is an area of bone, muscle and fat abundant in one selected slice. This is the tissue of interest that represents the various textures of the image that we analyzed in our previous work. In our medical practice, the shoulder is often diagnosed, which facilitates the acquisition of the required number of images.
Consider - Can body size play a role in this?
Body size may influence the parameters interplay chosen for the image creation especially repetition time where TR change interplay with signal to noise ratio. However this influence can be neglected as in our patients group we chose patients with similar BMI ratio - that information was included to the methods section. “In order to provide stable conditions for the images creation patients with BMI in the range of 20 – 26 were included to the study”
Consider - How can body size affect the results?
Body size may affect image creation as is usually associated proportionally with body mass increase and fat content. Above features may result in automatic increase of TR but even though reduction of signal to noise ratio in extreme cases is possible as SAR value (4 W/Kg of body mass) are not exceedable.
What about diff between males and females? Any thoughts?
We observed slight differences between male and female group. However, as group was set as consistent according to BMI also differences in sizes of the analyzed tissues were not significant.
Many figures – are all necessary? Consider to place figures side by side, to create a overview and comparison.
A figure related to phantom image analysis was removed as explained in one of the next responses.
We kept the figure structure and formatting unchanged because this will be processed and prepared by the JCM Editorial team. We hope that the final figure representation will the most suitable for their analysis.
Overall, what about the scan time – the scan time will always be affected when we adjust the scan parameters.
Scanning time is an important parameter in the diagnostic process and should be as short as possible. Firstly, for the patient's comfort, and secondly, to ensure smooth work flow in the diagnostic unit. However, the scanning time increases proportionally with the improved image quality. In the present work, the influence of the matrix size was analyzed - especially that this parameter directly and indirectly (by changing the FOV) influences the scanning time. Increasing the size of the matrix leads to the lengthening of the scanning protocol and this important parameter is discussed in section 3.3 of the Results section.
Discussion.
Why add phantom results/figure?
Did you scan patients and a phantom? The methods section does not include any phantom?
We scanned mostly patients, but also a phantom. However, this phantom study was performed only to demonstrate that correlation coefficients are very small and independent of the matrix size and noise for homogeneous phantom images (thus noise doesn’t influence much the correlation coefficients). This observation is rather obvious, and it was used just to explain why the correlation coefficients increase for larger matrices for noisy muscle texture. To conclude, even for such noisy images, bigger matrices are also recommended to ensure better image quality. Thus, the phantom image analysis is not very crucial for this study. Thus, the phantom results may be disregarded without affecting the conclusions presented regarding the effect of changing the matrix size on the properties of the muscle images.
Conclusion
Ok.
Overall interesting topic and paper.
Dear Reviewer, thank you for all your remarks, applied changes markedly improved our paper.
Reviewer 2 Report
The authors systematically investigated the effect of matrix sizes on texture analysis in MRI by evaluating the reproducibility of the texture features calculated in the muscle, fat, and bone on T2w images acquired with different matrix sizes. This topic is very interesting, and the authors drew several conclusions that could be helpful for radiologists and technologists to determine the optimal matrix size, and also be useful for further Radiomics studies. However, I have some minor comments here:
- The language can be further improved to be more clear and concise. Many sentences were ambiguous and difficult to follow (mainly in the Introduction and Discussion). I also recommend that the authors to further proofread the manuscript. For example, In the Introduction, “To our best knowledge influence of matrix size onto image quality was studied on the basis of.”, basis of what?
- In Methods, “A voxel of one plane non-isotropic resolution at 0.8x0.8x3 mm was acquired.” This resolution of 0.8x0.8 was only for the matrix size of 256x256 if I understood correctly. There should be multiple image resolution values in this study, and it’s more appropriate to provide a range.
- Please clarify how the ROIs were determined. Were they manually drawn by radiologists? Since each ROI was only a circle instead of covering a certain structure/organ, how was it determined on images with different matrix sizes? How did the authors make sure the ROIs could be consistent across all images?
- What is the gray level normalization used when calculating the texture features? Such information is important as it impacts the feature calculation.
- Many figures and data in the Discussion could be moved to Results as post-hoc analysis. Discussion should be focused on the interpretation of the results instead of providing new results.
Author Response
We would like to thank the Reviewer for valuable comments, that significantly improved the quality of this paper. All these comments were addressed in the paper text.
- The language can be further improved to be more clear and concise. Many sentences were ambiguous and difficult to follow (mainly in the Introduction and Discussion). I also recommend that the authors to further proofread the manuscript. For example, In the Introduction, “To our best knowledge influence of matrix size onto image quality was studied on the basis of.”, basis phantom with use of SNR values change as a reference
- In Methods, “A voxel of one plane non-isotropic resolution at 0.8x0.8x3 mm was acquired.” This resolution of 0.8x0.8 was only for the matrix size of 256x256 if I understood correctly. There should be multiple image resolution values in this study, and it’s more appropriate to provide a range.
Calculations were made for multiple voxel sizes obtained from different matrices. Voxel size for the different size was as follows . 0.8x0.8x3 for 256 × 256, 0.6x0.6x3 for 320 × 320, 0.5x0.5x3 for 384 × 384, 0.4x0.4x3 for 448 × 448 0.4x0.4x3 for 512 × 512.
- Please clarify how the ROIs were determined. Were they manually drawn by radiologists? Since each ROI was only a circle instead of covering a certain structure/organ, how was it determined on images with different matrix sizes? How did the authors make sure the ROIs could be consistent across all images?
The position of the ROI in the image was determined by the radiologist. The ROI size was calculated in proportion to the FOV and, consequently, to the image area and size of each tissue tested. The stability of the position and size of the ROI in the image was monitored in all examined cases. We added in section “Methods” paragraph related to this methodology.
- What is the gray level normalization used when calculating the texture features? Such information is important as it impacts the feature calculation.
The results of texture analysis depend on the image acquisition parameters. Different values of such parameters (like e.g., variable matrix size used in this study) may cause the brightness and contrast variation in individual regions of interest. As a result, the values of some texture features depend not only on the texture, but also on the ROIs brightness and/or contrast. For this reason, some of the features describe not only the structure of the tissue under examination but also the scanner's uneven sensitivity within the analyzed tissue region. This may in turn lead to an inappropriate description of the tissue. To limit these phenomena, the ROIs are normalized. Normalization is the stretching of the histogram in the ROI into the entire available intensity range. This improves the contrast of the investigated texture and reduces the influence of the ROI local mean intensity. Both effects improve the quality of the features obtained. One of the normalization methods is ±3σ normalization, the new range of intensities is defined as minnorm=μ-3σ and maxnorm=μ+3σ, where μ represents the mean intensity and σ denotes the standard deviation of the image intensities in the ROI. As a result, all original ROI intensities are mapped to this new intensity range. Such normalization is quite efficient in the case of classification of MR textures, as demonstrated in [28].
- Many figures and data in the Discussion could be moved to Results as post-hoc analysis. Discussion should be focused on the interpretation of the results instead of providing new results.
As recommended, Figs. 9 and 10 were moved to the results. In the Discussion, only 2 figures (11 and 12) remained as they explain the influence of SNR and image sharpness on image quality. These figures are used to develop recommendations for the implementation of the appropriate size of the matrix depending on the imaged tissue.